# Effects of Water Restriction and Supplementation on Cognitive Performances and Mood among Young Adults in Baoding, China: A Randomized Controlled Trial (RCT)

**DOI:** 10.3390/nu13103645

**Published:** 2021-10-18

**Authors:** Jianfen Zhang, Guansheng Ma, Songming Du, Shufang Liu, Na Zhang

**Affiliations:** 1Department of Nutrition and Food Hygiene, School of Public Health, Peking University, 38 Xueyuan Road, Haidian District, Beijing 100191, China; zjf@bjmu.edu.cn (J.Z.); mags@bjmu.edu.cn (G.M.); 2Laboratory of Toxicological Research and Risk Assessment for Food Safety, Peking University, 38 Xueyuan Road, Haidian District, Beijing 100191, China; 3Chinese Nutrition Society, 6 Guang An Men Nei Street, Xicheng District, Beijing 100053, China; dusm9709@126.com; 4School of Public Health, Hebei University Health Science Center, 342 Yuhua Road, Lianchi District, Baoding 071000, China; shufangliu@126.com

**Keywords:** water restriction, water supplementation, dehydration, rehydration, cognitive performances, mood

## Abstract

The brain is approximately 75% water. Therefore, insufficient water intake may affect the cognitive performance of humans. The present study aimed to investigate the effects of water restriction and supplementation on cognitive performances and mood, and the optimum amount of water to alleviate the detrimental effects of dehydration, among young adults. A randomized controlled trial was conducted with 76 young, healthy adults aged 18–23 years old from Baoding, China. After fasting overnight for 12 h, at 8:00 a.m. of day 2, the osmolality of the first morning urine and blood, cognitive performance, and mood were measured as a baseline test. After water restriction for 24 h, at 8:00 a.m. of day 3, the same indexes were measured as a dehydration test. Participants were randomly assigned into four groups: water supplementation group (WS group) 1, 2, or 3 (given 1000, 500, or 200 mL purified water), and the no water supplementation group (NW group). Furthermore, participants were instructed to drink all the water within 10 min. Ninety minutes later, the same measurements were performed as a rehydration test. Compared with the baseline test, participants were all in dehydration and their scores on the portrait memory test, vigor, and self-esteem decreased (34 vs. 27, *p* < 0.001; 11.8 vs. 9.2, *p* < 0.001; 7.8 vs. 6.4, *p* < 0.001). Fatigue and TMD (total mood disturbance) increased (3.6 vs. 4.8, *p* = 0.004; 95.7 vs. 101.8, *p* < 0.001) in the dehydration test. Significant interactions between time and volume were found in hydration status, fatigue, vigor, TMD, symbol search test, and operation span test (*F* = 6.302, *p* = 0.001; *F* = 3.118, *p* = 0.029; *F* = 2.849, *p* = 0.043; *F* = 2.859, *p* = 0.043; *F* = 3.463, *p* = 0.021) when comparing the rehydration and dehydration test. Furthermore, the hydration status was better in WS group 1 compared to WS group 2; the fatigue and TMD scores decreased, and the symbol search test and operation span test scores increased, only in WS group 1 and WS group 2 (*p* < 0.05). There was no significant difference between them (*p* > 0.05). Dehydration impaired episodic memory and mood. Water supplementation improved processing speed, working memory, and mood, and 1000 mL was the optimum volume.

## 1. Introduction

It is generally known that water takes up about 60–70% of the human body mass. It plays a crucial role in the processes of metabolism, such as temperature regulation and electrolyte balance maintenance [1]. The intake of water and the output of the water are balanced in our body. When the intake is more than the output, people may be in a state of hyperhydration, which may cause headaches, nausea, and even convulsions [2]. Moreover, insufficient input of water may place people into a state of dehydration. Dehydration may lead to headaches, urolithiasis, kidney dysfunction, physical decline, and gastrointestinal and cardiovascular function disorders [2,3,4,5,6]. Furthermore, because water is also the principal component of all cells, including those within the central nervous system, insufficient input of water may also influence cognitive performance.

Researchers have explored the effects of dehydration on cognitive performance. However, the results and conclusions have been inconsistent. Some aspects of cognitive performance, including vigilance and working memory among young males and females, were found to be impaired by mild dehydration [7,8]. In male basketball players, vigilance-related attention was impeded by dehydration [9]. Other studies, though, have found no negative effects on cognitive performance arising from dehydration [10,11]. Due to the different methodologies used to achieved dehydration, and different tests to evaluate cognitive performance, results in this research area have been inconsistent and may be confounded by various factors, including improvement induced by exercise and impairment induced by heat stress in cognitive performance [12,13,14,15]. Therefore, it is important to conduct studies using fluid deprivation to explore the effects of dehydration on aspects of cognitive performance. However, in the few research projects exploring the effects of dehydration on cognitive performance in which dehydration was elicited by fluid restriction, the conclusions have also been controversial. In a study conducted with young males and females, cognitive performance did not differ before and after 24 h of fluid deprivation [16]. However, in another study evaluating the effects of 24 h fluid deprivation among young healthy men, dehydration had detrimental effects on cognitive performance, including the total test solving time and total ballast time [17]. Even more important, in China, just one study has been carried out investigating the effects of fluid deprivation on the cognitive performance of adults. The study demonstrated that 36 h of fluids restriction exerted some adverse effects on the cognitive performances of the young adult male participants [18]; female young adults were not included in this study. Water intake is affected by many factors, including age, gender, and temperature, among others. Drinking patterns, including the amounts and contributions of the types of fluids, differ significantly among people in China, Europe, and the Americas. Therefore, although studies on hydration status and cognitive performance have been conducted in some countries, it is still necessary for China to undertake corresponding studies to obtain accurate local information on hydration status and cognitive performance.

Inconsistent effects of water supplementation or rehydration on cognitive performance have also been reported. Some studies have revealed no effect of water on cognitive performance; for both adults and children, water supplementation of 25 or 300 mL showed no improvement in visual attention [19]. Some researchers have hypothesized that cognitive performance could be improved by rehydration or water supplementation. Among children, there is some evidence to suggest that those with a water intake of an average of 409.1 mL, compared with those who drank nothing, had higher scores in visual attention and visual memory [20]. However, studies have generally obtained inconsistent conclusions [21,22]. In China, among young male adults, water supplementation of 1500 mL alleviated the adverse effects of dehydration on cognitive performance [18]. Nevertheless, in all the studies mentioned above, the most appropriate amount of water to address the adverse effects of dehydration on cognitive performance was not explored.

We hypothesized that dehydration after water deprivation for 36 h would have adverse effects on cognitive performance and mood, and that cognitive performance and mood could be improved by different amounts of water supplementation. Moreover, we hypothesized that the optimal amount of water to improve cognitive performance and mood would be 1000 mL.

The aims of the study were, firstly, to explore the effects of fluid restriction for 36 h on cognitive performance and mood; secondly, to investigate if water supplementation could improve cognitive performance and mood; and thirdly, to evaluate the appropriate amount of water for the maximum improvement of cognitive performance and mood after dehydration. This study will provide a more scientific basis for water intervention studies and health education and improve the understanding of the importance of adequate water intake.

## 2. Materials and Methods

### 2.1. Study Design

This was a randomized controlled, double-blinded trial study, which was implemented between 1st March 2018 and 31st May 2018 in Baoding, China. The registration number was ChiCTR-IOR-17011568. The study design is shown in Figure 1.

All participants were instructed to fast for 12 h. Participants were supplied with three meals, prepared by the study. They were divided into four groups randomly, which included the water supplementation groups (WS group 1, 2, 3) and the no water supplementation group (NW group). The amounts of water in WS groups 1, 2, and 3 were 1000, 500, and 200 mL, respectively. Cognitive performance and mood were measured three times. Measurements of anthropometric indexes (including height, weight, and blood pressure), blood biomarkers (including blood glucose and plasma osmolality), and urinary biomarkers (including osmolality and the specific gravity of urine) were executed at the same time.

### 2.2. Participants

The participants were young males and females recruited from one college in Hebei, China. The criteria of inclusion and exclusion are shown in Appendix A. The average age of the participants was 21 years old. The average height and weight of the participants were 166.3 cm and 62.1 kg, respectively.

### 2.3. Interventions

Participants were asked to undertake 12 h of water restriction, and their height, weight, blood pressure, and blood samples were obtained following standardized procedures for the baseline test. The first urine and fasting venous blood were also collected and tested. Mood, cognitive performance, and subjective sensation were determined with standard procedures by trained investigators. After that, participants were asked to undertake water restriction for 24 h, and urine samples were collected and sent to the laboratory of the hospital for measurements. Participants were randomly assigned to four groups, including WS group 1, WS group 2, WS group 3, and the NW group, with corresponding drinking water allocations of 1000, 500, 200, and 0 mL. After 90 min, the same information as outlined above was collected for the rehydration test.

### 2.4. Study Procedure

The study consisted of three stages over three days.

Stage 1: from 8:00 p.m. of day 1 to 8:00 a.m. of day 2. Participants were instructed to fast overnight for 12 h. However, before the protocol of the study (before 8:00 p.m. of day 1), participants were asked to maintain their fluid intakes as usual. They were also asked to sleep before 11:00 p.m. and not to urinate before getting up the next morning. At 8:00 a.m. of day 2, the baseline tests, including height, weight, blood pressure, and blood samples, were performed by the researchers, with urine samples and blood samples collected and tested. Furthermore, measurements of mood and cognitive performance, and the subjective sensation of thirst, were taken at the same time.

Stage 2: from 8:00 a.m. of day 2 to 8:00 a.m. of day 3. Participants undertook water restriction for 24 h under the instruction of the researchers. Three specific meals with a water content of ≤75% were supplied to them. In order to assess the amount of water from food, all the foods were weighed before and after the participants ate. Urine samples taken during the water restriction for 24 h were collected and measured.

Stage 3: from 8:00 a.m. of day 3 to 10:00 a.m. of day 3. Participants were given water supplementation. After arriving at the lab at 8:00 a.m. on day 3, they were instructed to take the dehydration test, with measurements of the related indexes. At 8:30 a.m., participants in the WS groups (WS groups 1, 2, and 3) were asked to drink 1000, 500, or 200 mL of purified water within 10 min, respectively, while the participants in the NW group did not drink any water or fluids. The water supplied to the participants was offered in three opaque containers. To reduce the gastrointestinal discomfort of the participants, the water was kept at 30–40 °C; investigators observed and recorded the temperature of the water carefully. Then, all participants were allowed to undertake normal activities under the observation of the researchers for 90 min. At 10:00 a.m., the rehydration test, including the same indexes as previously measured, was performed with standard procedures. The study procedure is shown in Figure 2.

During the study, in order to make sure that all the participants were healthy, they were asked if they were experiencing fever or other symptoms every day. The participants could quit at any time during the study. If this occurred, the researcher recorded the time and the reason they quit. During the three stages of the study, the participants were only allowed to undertake light activities, such as walking. During water supplementation, participants were closely watched in order to assess gastrointestinal discomfort. During this stage they could undertake activities in and around the laboratory, but long walks were not allowed.

### 2.5. Outcomes

All participants were assessed for the baseline test, dehydration test, and rehydration test using a questionnaire and software, completed face to face. The study outcomes are shown in Appendix A.

### 2.6. Randomization and Blinding

This was a randomized double-blinded study. The participants were identified by a unique number. Then, using a random number which was generated by the computer software, participants were assigned into the different groups. The investigators responsible for the statistical analysis were blinded to the collection and entry of the data. The water given to the participants was contained in three opaque cups of a certain weight, in order to disguise the amount of water in each cup. Furthermore, after water supplementation, the participants were not allowed to converse about the study.

### 2.7. Sample Size Calculation

The scores of the letter cancellation among participants were 23.34 and 28.81 before and after water supplementation [23]. The SAS 9.2 (SAS Institute Inc., Cary, NC, USA) was used to calculate the sample size. The power and α were set at 0.8 and 0.05, and a 10% drop-out rate was considered in the calculation. From this, we found 64 participants were needed in total.

### 2.8. Ethical Standards

The study protocol was approved by the Peking University Institutional Review Committee (IRB00001052-16071). The study was conducted according to the guidelines of the Declaration of Helsinki. Before participating in the study, all the participants signed the informed consent form.

### 2.9. Anthropometric Measurements

The anthropometry metrics, including height (H) and weight (W), were measured twice to the nearest 0.1 cm and 0.1 kg using a height–weight meter, with participants wearing light clothing and with bare feet (HDM-300, Huaju, Zhejiang, China). Blood pressure (BP) was measured twice to the nearest 2 mmHg, with at least a 15 min rest period before measurement by a professional nurse using an electronic sphygmomanometer (HEM-7051, Omron, Jiangsu, China). Two measurements were taken after 2 min intervals [24]. (BMI: weight (kg)/height squared (m)).

### 2.10. Temperature and Humidity of the Environment

The temperature and humidity of indoors and outdoors in the study locations were recorded for 3 days at three time points: 10:00 a.m., 2:00 p.m., and 8:00 p.m. each day (WSB-1-H2, Exasace, Zhengzhou, China).

### 2.11. Assessment of Total Water Intake (TWI)

During the 36 h of water restriction, participants had food supplied to them. All foods were weighed before and after the participants ate during day 2 (YP20001; SPC; Shanghai, China). The samples of the backup food were collected and sent to the laboratory of the Beijing Institute Nutritional Resource to measure the water content according to standard procedures [25].

### 2.12. Urinary and Plasma Biomarkers

Urine was collected by participants using containers designed by the researchers and was stored at +4 °C in the refrigerator before measurement. Blood samples were collected by trained nurses. After being centrifuged, all the blood samples were stored at −20 °C in the refrigerator before measurement. Urine and plasma osmolality were measured with the method of the freezing point using an osmotic pressure molar concentration meter (SMC 30C, Tianhe, Tianjin, China).

The electrolyte concentrations of sodium, potassium, chloride, calcium, magnesium, and phosphate in the urine and blood were assessed using an automatic biochemical analyzer (AU5800, Beckman, Brea, CA, USA) with the ion-selective electrode potentiometer method. Urine specific gravity (USG) was tested with the uric dry-chemistry method by a trained technician using an automatic urinary sediment analyzer (H-800, Dirui, Jilin, China).

Blood glucose (BG) was measured with an automatic biochemical analyzer (AU5800, Beckman, Brea, CA, USA), with the ion-selective electrode potentiometer method.

### 2.13. Hydration Status

The hydration status was defined using three categories, which included dehydration, middle hydration, and optimal hydration, defined by the osmolality of urine as greater than 800 mOsm/kg [26], between 500 mOsm/kg < urine osmolality ≤800 mOsm/kg, and ≤500 mOsm/kg [27,28], respectively.

### 2.14. Visual Analogue Scales (VAS) for Subjective Sensation

The questionnaire was designed to measure the subjective sensations of the participants, including satiety, hunger, sleepiness, and thirst, with four 10 cm lines. The left of the 10 cm line reflected the statement “not at all” and the right of the 10 cm line reflected the statement “extremely” [29,30,31]. Participants were asked to place pencil marks anywhere on the 10 cm line of the questionnaire. A higher score indicated a greater feeling of thirst and the other measured sensations.

### 2.15. Profile of Mood States (POMS)

The POMS was used to assess the mood states of the participants and included 7 subscales and 40 adjectives.

The seven subscales were tension/anxiety, depression/dejection, fatigue/inertia, vigor/activity, confusion/bewilderment, anger/hostility, esteem-related affect, and total mood disturbance (TMD). Participants were instructed to choose one scale for each adjective which conformed to their own situation from five scales, according to their mood state (0 not at all, 1 a little, 2 moderately, 3 quite a bit, 4 extremely). All answers were analyzed with a 5-point rating; (TMD = (tension + depression + anger + fatigue + confusion) − (vigor + esteem-related affect) + 100) [32].

### 2.16. Cognitive Performance (CP)

Cognitive performance was assessed with different cognitive tests, which were from “primary cognitive ability” software (Institute of Psychology, Chinese Academy Sciences, Beijing, China) [33].

### 2.17. Vocabulary Test

The test included 36 vocabulary words to evaluate the verbal comprehension of participants. Participants were instructed to choose the explanation that was the most accurate for the vocabulary word from five options, which included 1 most accurate option, 2 accurate options, and 2 wrong options. Participants were given scores of 2, 1, and 0 for choosing the most accurate, accurate, or wrong option, respectively. Higher scores demonstrated better verbal comprehension [34].

### 2.18. Similarities Test

The test had 32 vocabulary items to assess the verbal comprehension of participants. Participants were asked to give the similarities between the two words by choosing the most accurate explanation from five options, including 1 most accurate option, 2 accurate options, and 2 wrong options [34]. After participants chose the most accurate, accurate, or wrong option, they were given scores of 2, 1, and 0, respectively. The higher the score, the better their verbal comprehension.

### 2.19. Symbol Search Test

The symbol search test was used to assess the processing speed of the participants. In the middle of the computer screen, there was a line. Two symbols and five symbols were above and below the line, respectively. Participants were instructed to find the same symbol which occurred both above and below the line, as fast as possible, with a time limit of 120 s. Higher scores represented faster processing speed [34].

### 2.20. Operation Span Test

Working memory was evaluated using the operation span test. It consisted of a mental arithmetic task and a task remembering Chinese Zodiac Signs. Participants were asked to perform the mental arithmetic task and, at the same time, they were also asked to remember the Chinese Zodiac Signs that appeared following the mental arithmetic task. Moreover, participants were asked to finish each mental arithmetic task within a specified time (a few seconds). When they had finished all the mental arithmetic tasks, the order of the appearance of the Chinese Zodiac Signs was recalled by the participants. The scores of the test consisted of two components and the results were considered as valid scores only when the correct rate for the mental arithmetic task reached 80%. When the Chinese Zodiac Signs were recalled in the right order, the participants would achieve a score of 1. Higher scores represented better working memory [35].

### 2.21. Portrait Memory Test

The test consisted of six portraits to assess episodic memory. The six portraits were shown one by one, with the information about each person showed at the same time for a few seconds, including their last name, work, and a hobby. When the six portraits were shown for the first time, the participants were asked to recall all the information about the portrait appearing on the screen. After that, the six portraits appeared on the screen for a second time in a diverse order and the participants were asked to recall the related information again. When the participants chose the correct last name, work, and hobby, they obtained scores of 2, 1, and 1, respectively [33]. The higher the score, the better the person’s episodic memory was considered to be.

### 2.22. Statistical Analyses

All statistical analyses were run with SPSS Statistics 20.0 (IBM Corp., Armonk, NY, USA). Quantitative parameters of the participants, such as the age, height, and weight, were presented as the mean ± standard deviation. The differences between the four groups were explored using one-way classification ANOVA or Chi-square test for the baseline, dehydration, and rehydration tests. The paired-samples *t*-test and Chi-square test were used to explore the differences between the baseline and dehydration tests, and dehydration and rehydration tests, within the same group. The independent-samples *t*-test and Chi-square test were performed to find the differences between the groups. The hydration status, including dehydration, middle hydration, and optimal hydration, was compared in the four groups with the Chi-square test. The mixed model of repeated measurements (time and volume) was used to explore the effects of water supplementation on mood and cognitive performance, with the covaried indexes of age, gender, BMI, blood pressure, and blood glucose. Two-sided significance levels were set at 0.05 (*p* < 0.05), with 95% confidence intervals (95% CI).

## 3. Results

### 3.1. Characteristics of the Participants

A total of 76 participants, including 40 males and 36 females, were recruited and completed the study, with 20, 20, 18, and 18 participants (half males and hale females) in WS group 1, WS group 2, WS group 3, and the NW group, respectively, and with a 100% completion rate. The characteristics of the participants are shown in Table 1. There were no significant differences in the mean age, height, weight, BMI, and blood pressure among the four groups in the baseline, dehydration, and rehydration tests, respectively (*p* > 0.05). However, there were significant differences in weight, BMI, systolic pressure, and diastolic pressure when the baseline and dehydration test were compared (*p* < 0.05).

### 3.2. Temperature and Humidity

Table 2 shows that the average temperature and humidity indoors were 20.6 ± 0.1 °C and 60 ± 3%, respectively, and that of the outdoors were 20.7 ± 1.1 °C and 60 ± 5%, respectively.

### 3.3. Amounts of TWI

No significant differences in the amounts of water from food and the 24 h biomarkers, including the volume of urine, the osmolality, and USG, were found among the four groups during the water restriction period of 24 h (*p* > 0.05), as shown in Table 3. The average amount of water from food, the voids of urine, and the volume of urine of the participants was 920, 5, and 900 mL, respectively.

### 3.4. Thirst, Urinary, and Plasma Biomarkers

Compared with the baseline test scores for thirst, the osmolality of urine, the concentrations of Na, Cl, Ca, and Mg, the USG of urine, and the osmolality of plasma, the concentrations of Mg, creatine, and nitrogen of plasma were significantly higher in the dehydration test. In terms of hydration status, all participants were in the state of dehydration (*p* < 0.05) during the dehydration test, as shown in Table 4.

Comparing the rehydration and dehydration test, a significant interaction between time and volume was found for thirst (*p* < 0.05). Furthermore, there were significant reductions in thirst for those in WS group 1, 2, and 3 (*p* < 0.001; *p* < 0.001; *p* = 0.018), with decreases of 4.7, 2.9, and 1.5, respectively. There was an increase of 0.4 in the NW group, but this was not significant (*p* = 0.259). The improvement of thirst was greater in WS group 1 than WS group 2 (*p* = 0.004), but no significant differences were found between WS group 2 and WS group 3 (*p* = 0.104). In the urine biomarkers, a significant interaction between time and volume was found in the osmolality (*p* < 0.05); moreover, the osmolality decreased significantly in the rehydration test compared to the dehydration test in WS group 1, WS group 2, and WS group 3 (*p* < 0.001; *p* = 0.025; *p* = 0.021), with reductions of 658, 191, and 172 mOsm/kg, respectively. There was a significant increase of 39 mOsm/kg in the NW group (*p* = 0.010). Moreover, the osmolality of urine in WS group 1 was lower than in WS group 2 (*p* < 0.001) and no significant difference was found between WS group 2 and WS group 3 (*p* = 0.747).

As for the hydration status, the statuses of nine and five participants changed from dehydration to optimal hydration in WS group 1 and WS group 2, respectively (*p* < 0.001; *p* = 0.047). In WS group 3, no significant difference in the number of participants was found between the dehydration and rehydration tests, even though the statuses of two participants changed (*p* = 0.104). In the NW group, there was no significant difference in the hydration status (*p* = 1.000). When comparing WS group 1 and WS group 2, the hydration status was better in WS group 1 (*p* < 0.001) in the rehydration test.

As for the plasma biomarkers, no significant interactions between time and volume were found, including the osmolality (all *p* > 0.05), as shown in Table 5.

### 3.5. Effects of Dehydration and Rehydration on POMS

Table 6 shows that the scores for fatigue and TMD were higher (*p* < 0.005) and the scores for vigor and esteem-related affect decreased significantly (all *p* < 0.001) when comparing the baseline test with the dehydration test. Furthermore, comparing the rehydration with dehydration tests, significant interactions between time and volume were found in fatigue, vigor, and TMD (*p* < 0.05). As for fatigue, significant decreases of 1.3, 2.5, and 1.8 were found in WS group 1, WS group 2, and WS group 3, respectively (*p* = 0.049; *p* = 0.038; *p* = 0.047), with an increase of 1.4 in the NW group (*p* = 0.041) when comparing the rehydration test with the dehydration test. In terms of vigor, there were no significant increases in WS group 1 and WS group 2 (*p* = 0.398; *p* = 0.735), but a significant increase was found in WS group 3 (*p* = 0.045), with no significant decrease in the NW group (*p* = 0.052). As for TMD, significant decreases were found in WS group 1 and WS group 2 (*p* = 0.034; *p* = 0.047), but no significant differences were found in WS group 3 and the NW group (*p* = 0.057; *p* = 0.375). Moreover, no significant differences were found in fatigue and TMD between WS group 1 and WS group 2 (*p* = 0.648; *p* = 0.267).

### 3.6. Effects of Dehydration and Rehydration on Cognitive Performance (CP)

Table 7 shows significant increases in the symbol search test and operation span test (*p* < 0.05), but a decrease in the portrait memory test (*p* < 0.05), when comparing the dehydration test with the baseline test. Significant interactions between TIME and VOLUME were found in the symbol search test and operation span test (*p* < 0.05) when comparing the rehydration test with the dehydration test. Furthermore, in the symbol search test, the scores increased significantly in WS group 1 and WS group 2 (*p* = 0.003; *p* = 0.005), with no significant changes in WS group 3 and the NW group (*p* = 0.109; *p* = 0.572). Moreover, the symbol search test scores did not differ significantly between WS group 1 and WS group 2 (*p* = 0.912). As for the operation span test, significant increases were found in WS group 1 and WS group 2 (*p* = 0.009; *p* = 0.005), with no significant changes in WS group 3 and the NW group (*p* = 0.589; *p* = 0.410). Comparing WS group 1 with WS group 2, no significant differences were found in the operation span test (*p* = 0.512). In the portrait memory test, there was no significant interaction between time and volume (*p* > 0.05), with only an increase in WS group 1 (*p* = 0.049), no significant increases in WS group 2 and WS group 3 (*p* = 0.654; *p* = 0.179), and no significant decrease in the NW group (*p* = 0.476).

**Table 4 nutrients-13-03645-t004:** The thirst and urine biomarkers of the participants.

	Baseline Test					Dehydration Test					*t*	*p*	Rehydration Test				Interaction	
	WS Group 1 (*n* = 20)	WS Group 2 (*n* = 20)	WS Group 3 (*n* = 18)	NW Group (*n* = 18)	Total	WS Group 1 (*n* = 20)	WS Group 2 (*n* = 20)	WS Group 3 (*n* = 18)	NW Group (*n* = 18)	Total			WS Group 1 (*n* = 20)	WS Group 2 (*n* = 20)	WS Group 3 (*n* = 18)	NW Group (*n* = 18)	*F*	*p*
Thirst	5.2 ± 2.1	4.7 ± 2.5	4.3 ± 2.6	4.5 ± 2.7	4.6 ± 2.3	6.4 ± 2.1	6.4 ± 2.1	6.2 ± 2.8	6.0 ± 2.4	6.3 ± 2.3	−6.758	<0.001	1.7 ± 1.7 ^#†^	3.5 ± 2.1 ^†^	4.7 ± 2.4 ^†^	6.4 ± 2.1	19.847	<0.001
Urinary Biomarkers																		
Osmolality (mOsm/kg) *	840 ± 261	829 ± 170	786 ± 205	830 ± 234	822 ± 217	1111 ± 130	1079 ± 114	1093 ± 71	1092 ± 78	1094 ± 101	−11.221	<0.001	453 ± 273 ^#†^	888 ± 352 ^†^	921 ± 265 ^†^	1131 ± 72	23.697	<0.001
Na (mmol/L) *	167 ± 64	167 ± 69	170 ± 60	160 ± 61	166 ± 62	228 ± 44	225 ± 37	254 ± 31	241 ± 37	236 ± 39	−8.784	<0.001	72 ± 55 ^#^	151 ± 65	195 ± 87	199 ± 59	9.875	<0.001
K (mmol/L)	43.3 ± 21.7	42.6 ± 20.2	40.3 ± 22.0	50.5 ± 22.6	44.1 ± 21.5	46.3 ± 23.7	44.4 ± 17.0	40.5 ± 19.3	42.5 ± 16.6	43.5 ± 19.1	0.219	0.828	34.1 ± 23.0 ^#^	67.8 ± 35.0	58.3 ± 21.6	83.5 ± 21.1	11.986	<0.001
Cl (mmol/L) *	148 ± 52	156 ± 55	153 ± 58	154 ± 58	153 ± 55	196 ± 46	196 ± 45	210 ± 43	202 ± 38	201 ± 43	−6.029	<0.001	91 ± 67 ^#^	187 ± 80	218 ± 87	253 ± 71	11.354	<0.001
Ca (mmol/L) *	2.61 ± 1.25	2.96 ± 1.84	2.38 ± 1.44	2.74 ± 1.86	2.68 ± 1.60	7.20 ± 3.48	6.84 ± 3.19	7.02 ± 2.57	6.76 ± 2.65	6.96 ± 2.96	−13.321	<0.001	1.54 ± 1.58 ^#^	2.81 ± 1.59	2.94 ± 1.69	3.71 ± 1.05	2.617	0.057
Phosphorus (mmol/L)	39.94 ± 19.50	32.85 ± 10.93	36.90 ± 16.10	37.08 ± 15.58	36.67 ± 15.72	43.01 ± 17.41	40.08 ± 11.62	36.63 ± 10.78	37.17 ± 11.98	39.35 ± 13.28	−1.398	0.166	9.64 ± 6.93 ^#^	23.30 ± 12.75	18.34 ± 7.46	27.79 ± 10.87	7.916	<0.001
Mg (mmol/L) *	3.49 ± 1.49	3.00 ± 1.41	3.10 ± 1.15	3.77 ± 2.60	3.33 ± 1.73	7.07 ± 3.20	6.03 ± 3.36	7.58 ± 3.53	6.44 ± 3.56	6.77 ± 3.39	−8.457	<0.001	1.41 ± 0.90 ^#^	2.62 ± 1.55	3.05 ± 1.37	3.83 ± 1.58	3.063	0.033
pH	6.38 ± 0.36	6.35 ± 0.37	6.39 ± 0.37	6.36 ± 0.33	6.37 ± 0.35	6.40 ± 0.42	6.30 ± 0.34	6.39 ± 0.21	6.44 ± 0.29	6.38 ± 0.33	−0.293	0.770	6.1 ± 0.3	6.2 ± 0.4	6.1 ± 0.2	6.1 ± 0.2	1.041	0.380
USG	1.025 ± 0.006	1.025 ± 0.004	1.024 ± 0.006	1.026 ± 0.005	1.025 ± 0.005	1.028 ± 0.004	1.029 ± 0.003	1.029 ± 0.004	1.029 ± 0.002	1.029 ± 0.003	−5.019	<0.001	1.014 ± 0.009 ^†#^	1.025 ± 0.008	1.026 ± 0.007	1.029 ± 0.004	13.475	<0.001
Hydration statuses (%) *																		
Dehydration	13 (65.0)	12 (60.0)	9 (50.0)	10 (55.6)		20 (100.0)	20 (100.0)	18 (100.0)	18 (100.0)				3 (15.0) ^#†^	15 (75.0) ^†^	14 (77.8)	18 (100.0)	35.714	<0.001
Middle hydration	5 (25.0)	8 (40.0)	8 (44.4)	5 (27.8)		0 (0.0)	0 (0.0)	0 (0.0)	0 (0.0)		46.973	<0.001	4 (20.0)	0 (0.0)	2 (11.1)	0 (0.0)		
Optimal hydration	2 (10.0)	0 (0.0)	1 (5.6)	3 (16.7)		0 (0.0)	0 (0.0)	0 (0.0)	0 (0.0)				13 (65.0)	5 (25.0)	2 (11.1)	0 (0.0)		

Note: Values are shown as the mean ± standard deviation (SD), with the exception that percentages are shown as *n* (percentage); *, there were significant differences between the baseline and dehydration test; ^#^, there were significant differences between the four groups; ^†^, there were significant differences between dehydration test and rehydration test within the group. Compared with the baseline test, the thirst, the osmolality of urine, and USG increased (*t* = −6.758, *p <* 0.001; *t* = −11.221, *p <* 0.001; *t* = −5.019, *p* < 0.001), significant differences were found in the hydration statuses (*F* = 46.973, *p* < 0.001); the osmolality of plasma increased (*t* = −6.157, *p <* 0.001) in the dehydration test. Compared with the dehydration test, in the rehydration test there were significant decreases in WS group 1, WS group 2, and WS group 3 in thirst (*t* = 8.661, *p* < 0.001; *t* = 6.099, *p* < 0.001; *t* = 2.605, *p* = 0.018). No significant increase was found in the NW group (*t* = −1.169, *p* = 0.259). The osmolality of urine in WS group 1, WS group 2, and WS group 3 decreased (*t* = 11.247, *p* < 0.001; *t* = 2.440, *p* = 0.025; *t* = 2.533, *p* = 0.021), but there was a significant increase in the NW group (*t* = −2.886, *p* = 0.010). In hydration status, significant differences were found in WS group 1 and WS group 2 (χ^2^ = 32.039, *p* < 0.001; χ^2^ = 7.648, *p* = 0.047), while no significant differences were found in WS group 3 and the NW group (χ^2^ = 3.743, *p* = 0.104; *p* = 1.000). When comparing thirst during the rehydration test, the improvement in thirst in WS group 1 was greater than in WS group 2 (*t* = −3.204, *p* = 0.004), but no significant difference was found between WS group 2 and WS group 3 (*t* = −1.666, *p* = 0.104). In the osmolality of urine, WS group 1 was lower than WS group 2 (*t* = −4.365, *p* < 0.001), while no significant difference was found between WS group 2 and WS group 3 (*t* = −0.324, *p* = 0.747). In hydration status, WS group 1 was better than WS group 2 (χ^2^ = 15.397, *p* < 0.001).

**Table 5 nutrients-13-03645-t005:** The plasma biomarkers of the participants.

	Baseline Test					Dehydration Test					*t*	*p*	Rehydration Test				Interaction	
	WS Group 1 (*n* = 20)	WS Group 2 (*n* = 20)	WS GROUP 3 (*n* = 18)	NW group (*n* = 18)	Total	WS Group 1 (*n* = 20)	WS Group 2 (*n* = 20)	WS Group 3 (*n* = 18)	NW Group (*n* = 18)	Total			WS Group 1 (*n* = 20)	WS Group 2 (*n* = 20)	WS Group 3 (*n* = 18)	NW Group (*n* = 18)	*F*	*p*
Osmolality (mOsm/kg) *	293 ± 5	293 ± 7	294 ± 5	293 ± 4	293 ± 5	297 ± 6	298 ± 4	296 ± 5	296 ± 4	297 ± 5	−6.157	<0.001	292 ± 4 ^†^	294 ± 4	293 ± 5	295 ± 6	2.035	0.117
Na (mmol/L)	140 ± 2	140 ± 1	140 ± 1	140 ± 2	140 ± 1	140 ± 1	140 ± 2	141 ± 1	138 ± 11	140 ± 6	0.716	0.476	138 ± 2	139 ± 2	139 ± 2	140 ± 2	2.128	0.104
K (mmol/L)	4.4 ± 0.6	4.5 ± 0.6	4.6 ± 1.0	4.6 ± 0.6	4.51 ± 0.70	4.4 ± 0.4	4.5 ± 0.4	4.4 ± 0.4	4.5 ± 0.5	4.43 ± 0.40	0.842	0.403	4.2 ± 0.3	4.3 ± 0.3	4.3 ± 0.3	4.5 ± 0.4	0.796	0.500
Cl (mmol/L)	105 ± 2	104 ± 2	105 ± 1	104 ± 2	104 ± 2	104 ± 2	105 ± 1	105 ± 2	103 ± 8	104 ± 4	0.844	0.402	102 ± 2^#^	103 ± 1	104 ± 2	104 ± 3	2.816	0.045
Ca (mmol/L)	2.41 ± 0.08	2.40 ± 0.08	2.42 ± 0.08	2.42 ± 0.07	2.41 ± 0.08	2.45 ± 0.08	2.42 ± 0.08	2.46 ± 0.09	2.44 ± 0.09	2.44 ± 0.08	−3.477	0.001	2.47 ± 0.08	2.46 ± 0.06	2.48 ± 0.09	2.47 ± 0.08	0.647	0.588
Phosphorus (mmol/L)	1.30 ± 0.17	1.30 ± 0.21	1.28 ± 0.14	1.32 ± 0.15	1.30 ± 0.17	1.37 ± 0.16	1.30 ± 0.20	1.26 ± 0.12	1.32 ± 0.15	1.31 ± 0.16	−0.657	0.513	1.25 ± 0.07	1.21 ± 0.13	1.16 ± 0.14	1.24 ± 0.09	0.366	0.778
Mg (mmol/L)	0.88 ± 0.06	0.88 ± 0.05	0.90 ± 0.07	0.89 ± 0.05	0.89 ± 0.06	0.86 ± 0.05	0.87 ± 0.06	0.89 ± 0.06	0.89 ± 0.04	0.88 ± 0.05	1.691	0.095	0.83 ± 0.06^#^	0.86 ± 0.06	0.87 ± 0.04	0.89 ± 0.05	3.071	0.033
Creatine (mmol/L) *	62 ± 13	61 ± 11	66 ± 15	66 ± 10	63 ± 12	62 ± 12	61 ± 12	65 ± 15	63 ± 12	62 ± 13	2.803	0.006	58 ± 13	59 ± 13	62 ± 17	60 ± 10	0.822	0.486
Nitrogen (mmol/L)*	4.41 ± 1.09	4.53 ± 1.31	4.54 ± 0.99	4.09 ± 1.19	4.40 ± 1.14	6.63 ± 1.39	6.58 ± 1.45	6.34 ± 0.98	5.91 ± 1.06	6.38 ± 1.25	−20.576	<0.001	5.76 ± 1.34	6.03 ± 1.43	5.85 ± 1.03	5.76 ± 1.08	8.878	<0.001

Note: Values are shown as the mean ± standard deviation (SD); *, there were significant differences between the baseline and dehydration test; ^#^, there were significant differences between the four groups; ^†^, there were significant differences between the dehydration test and rehydration test within the group. Compared with the baseline test, the osmolality, the concentrations of Ca, creatine, and nitrogen all increased (*t* = −6.157, *p* < 0.001; *t* = −3.477, *p* = 0.001; *t* = 2.803, *p* = 0.006; *t* = −20.576, *p* < 0.001) in the dehydration test.

**Table 6 nutrients-13-03645-t006:** The POMS of the participants.

	Baseline Test					Dehydration Test					*t*	*p*	Rehydration Test				Interaction	
	WS Group 1 (*n* = 20)	WS Group 2 (*n* = 20)	WS Group 3 (*n* = 18)	NW Group (*n* = 18)	Total	WS Group 1 (*n* = 20)	WS Group 2 (*n* = 20)	WS Group 3 (*n* = 18)	NW Group (*n* = 18)	Total			WS Group 1 (*n* = 20)	WS Group 2 (*n* = 20)	WS Group 3 (*n* = 18)	NW Group (*n* = 18)	*F*	*p*
Tension	3.8 ± 2.5	3.4 ± 2.4	4.1 ± 3.5	3.7 ± 2.3	3.8 ± 2.7	3.7 ± 3.3	3.1 ± 3.0	4.3 ± 5.0	4.2 ± 3.7	3.8 ± 3.8	−0.146	0.885	1.8 ± 2.2 ^†^	2.0 ± 2.9 ^†^	2.4 ± 3.4	3.8 ± 4.4	1.043	0.379
Anger	1.8 ± 2.9	2.5 ± 3.6	2.5 ± 3.0	1.9 ± 1.9	2.2 ± 2.9	2.1 ± 3.6	2.1 ± 3.0	2.7 ± 3.7	2.7 ± 3.8	2.4 ± 3.5	−0.579	0.564	0.9 ± 2.2 ^†^	2.0 ± 3.4	1.9 ± 3.6	2.4 ± 3.7	0.663	0.578
Fatigue *	3.2 ± 2.9	4.3 ± 4.1	4.3 ± 2.6	2.8 ± 2.2	3.6 ± 3.1	4.8 ± 3.9	4.9 ± 4.8	4.7 ± 3.5	4.8 ± 4.7	4.8 ± 4.2	−2.976	0.004	3.1 ± 2.4 ^†^	3.5 ± 3.1 ^†^	4.1 ± 3.2	6.9 ± 4.3 ^†^	6.302	0.001
Depression	1.8 ± 2.3	2.4 ± 2.6	2.4 ± 3.4	2.7 ± 2.8	2.3 ± 2.7	2.5 ± 3.5	2.5 ± 3.4	3.1 ± 3.7	3.1 ± 4.1	2.8 ± 3.6	−1.325	0.189	1.3 ± 2.3 ^†^	2.5 ± 3.6	1.8 ± 3.7 ^†^	2.8 ± 3.5	1.673	0.180
Confusion	3.2 ± 2.0	3.6 ± 2.3	4.1 ± 3.2	3.3 ± 2.1	3.5 ± 2.4	3.3 ± 2.8	3.6 ± 3.3	3.9 ± 3.4	3.9 ± 2.8	3.7 ± 3.0	−0.547	0.586	2.5 ± 2.4	2.8 ± 3.3	2.4 ± 3.4 ^†^	3.5 ± 2.7	0.638	0.593
Vigor *	12.9 ± 4.9	10.4 ± 4.7	11.7 ± 4.0	12.2 ± 3.2	11.8 ± 4.3	11.0 ± 5.5	8.9 ± 4.4	7.6 ± 6.0	9.4 ± 4.4	9.2 ± 5.2	6.479	<0.001	11.3 ± 5.3	9.2 ± 4.1	9.6 ± 4.6 ^†^	7.7 ± 3.8	3.188	0.029
Esteem-related affect *	8.2 ± 3.2	7.4 ± 3.1	7.9 ± 2.9	7.9 ± 2.0	7.8 ± 2.8	7.1 ± 3.8	6.6 ± 3.0	5.8 ± 3.2	5.9 ± 3.3	6.4 ± 3.3	5.828	<0.001	6.8 ± 3.6	6.7 ± 3.2	5.9 ± 3.3	5.7 ± 3.4	0.237	0.871
TMD *	92.6 ± 13.8	98.3 ± 15.6	97.8 ± 14.8	94.2 ± 11.2	95.7 ± 13.9	98.5 ± 19.0	100.9 ± 17.1	105.4 ± 22.0	103.4 ± 21.0	101.8 ± 19.6	−3.699	<0.001	91.9 ± 12.7 ^†^	96.9 ± 17.3 ^†^	97.2 ± 19.1	106.7 ± 20.1	2.849	0.043

Note: Values are shown as the mean ± standard deviation (SD); *, there were significant differences between the baseline and dehydration test; ^†^, there were significant differences between the dehydration test and rehydration test within the group. Compared with the dehydration test, for fatigue, in the rehydration test there were significant decreases of scores in WS group 1, WS group 2, and WS group 3 (*t* = 2.100, *p* = 0.049; *t* = 2.236, *p* = 0.038; *t* = 2.138, *p* = 0.047), and a significant increase in the NW group (*t* = −2.215, *p* = 0.041). In vigor, there were no significant increases in WS group 1 and WS group 2 (*t* = −0.865, *p* = 0.398; t = −0.343, *p* = 0.735), but the increase in WS group 3 was significant (*t* = −2.161, *p* = 0.045), with no significant decrease in the NW group (t = 2.093, *p* = 0.052). In TMD, significant decreases were found in WS group 1 and WS group (*t* = 2.283, *p* = 0.034; *t* = 2.127, *p* = 0.047), and no significant differences were found in WS group 3 and the NW group (*t* = 2.039, *p* = 0.057; *t* = −0.912, *p* = 0.375). Comparing WS group 1 with WS group 2, no significant differences were found in fatigue, vigor, and TMD (*t* = −0.460, *p* = 0.648; *t* = 1.442, *p* = 0.157; *t* = −1.127, *p* = 0.267).

**Table 7 nutrients-13-03645-t007:** The CP of participants.

	Baseline Test					Dehydration Test					*t*	*p*	Rehydration Test				Interaction	
	WS Group 1 (*n* = 20)	WS Group 2 (*n* = 20)	WS Group 3 (*n* = 18)	NW Group (*n* = 18)	Total	WS Group 1 (*n* = 20)	WS Group 2 (*n* = 20)	WS Group 3 (*n* = 18)	NW Group (*n* = 18)	Total			WS Group 1 (*n* = 20)	WS Group 2 (*n* = 20)	WS Group 3 (*n* = 18)	NW Group (*n* = 18)	*F*	*p*
Vocabulary test	55 ± 4	56 ± 7	58 ± 7	57 ± 3	57 ± 6	55 ± 4	56 ± 7	58 ± 7	57 ± 3	57 ± 6	-	-	55 ± 4	56 ± 7	58 ± 7	57 ± 3	-	-
Similarities test	50 ± 8	49 ± 7	48 ± 8	51 ± 7	50 ± 7	50 ± 8	49 ± 7	48 ± 8	51 ± 7	50 ± 7	-	-	50 ± 8	49 ± 7	48 ± 8	51 ± 7	-	-
Symbol search test *	37 ± 6	39 ± 5	39 ± 7	36 ± 5	38 ± 6	43 ± 6	44 ± 3	45 ± 8	42 ± 8	43 ± 6	−8.956	<0.001	45 ± 6 ^†^	47 ± 5 ^†^	47 ± 9	41 ± 7	2.859	0.043
Operation span test *	9 ± 3	9 ± 2	7 ± 4	8 ± 3	8 ± 3	10 ± 2	10 ± 3	8 ± 4	9 ± 3	9 ± 3	−3.909	<0.001	11 ± 2 ^†^	11 ± 3 ^†^	10 ± 3	9 ± 3	3.463	0.021
Portrait memory test *	33 ± 9	37 ± 7	32 ± 7	34 ± 8	34 ± 8	25 ± 8	27 ± 9	25 ± 11	29 ± 10	27 ± 10	6.782	<0.001	29 ± 7 ^†^	28 ± 10	27 ± 10	28 ± 7	1.496	0.223

Note: Values are shown as the mean ± standard deviation (SD); *, there were significant differences between the baseline and dehydration test; ^†^, there were significant differences between the dehydration test and rehydration test within the group. Comparing the dehydration test with the rehydration test, in the symbol search test, significant increases were found in WS group 1 and WS group 2 (*t* = −3.410, *p* = 0.003; *t* = −3.138, *p* = 0.005), but no significant changes were found in WS group 3 and the NW group (*t* = −1.694, p = 0.109; *t* = 0.576, *p* = 0.572). Moreover, no significant differences were found in the symbol search test between WS group 1 and WS group 2 (*t* = 0.111, *p* = 0.912). As for the operation span test, significant increases were found in WS group 1 and WS group 2 (t = −2.896, *p* = 0.009; *t* = −3.176, *p* = 0.005), but no significant changes were found in WS group 3 and the NW group (*t* = −0.551, *p* = 0.589; *t* = 0.844, *p* = 0.410). Comparing WS group 1 and WS group 2, no significant differences were found in the operation span test (*t* = 0.661, *p* = 0.512). In the portrait memory test, there was no significant interaction (*F* = 1.496, *p* = 0.223), with an increase in WS group 1 (*t* = −2.105, *p* = 0.049), no significant increases in WS group 2 and WS group 3 (*t* = −0.455, *p* = 0.654; *t* = −1.403, *p* = 0.179), and no significant decrease in the NW group (*t* = 0.730, *p* = 0.476).

## 4. Discussion

In the present study, a double-blind randomized controlled trial was implemented to explore the effects of water restriction and water supplementation on the cognitive performance of young males and females. Moreover, the study also aimed to find the most appropriate volume of water to ameliorate the adverse effects of dehydration on cognitive performance and mood to the greatest extent.

Studies have shown that habitual caffeine consumption influences aspects of cognition, such as attention and memory [36,37]. In our study, none of the participants engaged in habitual caffeine consumption. Furthermore, in each group, the number of males and females was the same and the groups did not differ significantly in the proportion of males and females (χ^2^ = 0.000, *p* = 1.000).

Compared with water restriction for 12 h, the osmolality of urine, USG, and thirst increased after water restriction for 36 h, which indicated that a prolonged period of water restriction can cause more serious negative effects due to dehydration. After water restriction for 36 h, the osmolality of urine was higher than 800 mOsm/kg, and all the participants were in dehydration in our study. The results were similar to a study conducted with 16 males and females, which showed that the osmolality of urine and thirst were higher after water deprivation for 28 h compared with deprivation for 24 h [16]. In our study, after water supplementation, thirst decreased in participants who drank 1000, 500, or 200 mL. Moreover, the higher the volume of water the participants drank, the lower their scores for thirst. This meant that the most appropriate volume of water to reduce thirst was 1000 mL. Similarly, after water supplementation, the hydration status of participants who drank 1000 or 500 mL of water was better than those who drank 200 mL or no water after water restriction for 36 h. Meanwhile, the hydration status of participants given 1000 mL water was better than participants given 500 mL water. Therefore, the appropriate volume of water to improve the hydration status of participants after water deprivation of 36 h was 1000 mL.

In our study, water restriction for 36 h had a negative impact on vigor and esteem-related affect, which was consistent with the results of some previous studies. In one study implemented with 24 females, after water restriction for 24 h, alertness was reduced, while tiredness, fatigue, and confusion were increased [38]. Another study conducted with young male college students found vigor and esteem-related affect were impaired and TMD and fatigue increased [18]. Moreover, in another study of young males and females, after water deprivation for 28 h, fatigue increased and alertness decreased [16]. Furthermore, a study implemented with 32 men showed that TMD, fatigue, confusion, anger, and depression increased [39]. As mentioned above, one aim of this study was to explore whether water supplementation could improve mood. The results showed that water supplementation alleviated fatigue and TMD in our study. Comparisons showed that the most appropriate volume of water to improve mood after water restriction for 36 h was 500 mL or 1000 mL. A self-controlled study of 12 young males demonstrated that water supplementation of 1500 mL improved fatigue and TMD [18]. Despite the research mentioned above, the results from other studies have differed. According to a study conducted with 10 well-trained athletes, alertness did not change during rehydration [40]. Similarly, a study implemented with 29 young adults revealed that the mood scale was unaffected by water supplementation of 303.44 mL [41]. The differences in the questionnaires used to evaluate mood states in the studies may have contributed to differences in results.

For cognitive performance, compared to water restriction for 12 h, the scores of the symbol search test and operation span test were higher after water deprivation for 36 h. This indicated that practice may improve the scores of cognitive tests. A previous study demonstrated that practice also improved the scores of cognitive performances among 34 adults [42]. For school children, improvements in cognitive performance have also been related to practice [43]. Contrary to the results mentioned above, adverse effects of dehydration on working memory were found in some studies. In a study of young male adults, it was revealed that working memory was impaired after exercise in high temperatures [7]. Another randomized controlled study showed that after exposure to a temperature of 30 °C for 4 h, the working memory of 118 adults was attenuated [44]. The scores of the portrait memory test were impeded in our study, which indicated that episodic memory (long-term memory) was impaired after 36 h of water deprivation. Similar to the results of a randomized controlled trail conducted with seven adults, significant deterioration was observed in episodic memory [45]. Furthermore, a meta-analysis demonstrated that dehydration impedes cognitive performance, in particular, executive function and attention [46]. There have been conflicting findings about the effects of dehydration on cognitive performance and some studies did not find obvious impairments. After water deprivation for 28 h, the cognitive performance of participants was not altered [16]. Moreover, neither dehydration nor hyperthermia impaired the cognitive performance in the memory or perceptual domains among eight young healthy males [47]. As previously mentioned, the main focus of this study was to discover whether water supplementation or rehydration could improve cognitive performance, and which volume of water would be the most appropriate amount to amend the detrimental effects of dehydration. We found that water supplementation improved cognitive performance, including processing speed and working memory, but the effects of 500 mL and 1000 mL were not significantly different, similar to the findings of another study [44]. One study showed that visual and working memory and executive function were improved after rehydration in a sample of 12 women (participants drank 2500 mL/24 h) [8]. Moreover, a significant improvement after water supplementation of 167 mL in an attention task was found in a sample of 44 adults [23]. After drinking 500 mL of water, better scores in judgement and decision-making tasks were observed for 29 adults [42]. Among children aged 5–12 years, after water supplementation of 750 mL, better scores in long-term attention and working memory were obtained [22]. Moreover, in children in 5th and 6th grade in Germany, the consumption of water of up to 1000 mL (or up to 50% of TWI) led to better cognitive performance than children who drank less water in the control group, especially in short-term memory [48]. Nevertheless, the results of some studies have been inconsistent with the studies mentioned above. In both adults and children, no improvement in cognitive performance was found after water replenishment of 900 mL in two studies [7,49]. A systematic review concluded that the benefits of water supplementation on cognitive performance were not apparent, and more studies are needed [50]. Episodic memory was not improved in our study after water supplementation and a greater volume of water may be needed to remit the impact of dehydration on episodic memory. Therefore, we concluded that the most appropriate volume of water to improve cognitive performance after water restriction for 36 h is 500 or 1000 mL.

Our study had both strengths and weaknesses. Firstly, a double-blind randomized controlled trial was conducted to reduce bias, with the investigators conducting the measurements, the data analysts, and the participants all blinded to the intervention protocol. Secondly, in order to make sure that the participants adhered to the protocols, the hydration status of the participants was measured with the indexes of USG and the osmolality of urine and plasma. Thirdly, the water temperature was maintained at 30–40 °C, which was suitable for drinking, so as to reduce the gastrointestinal response of the participants. Regarding the weaknesses of our study, the impacts of long-term water supplementation on cognitive performance were not explored. Furthermore, the effects of different water temperatures on cognitive performance and mood were not studied.

## 5. Conclusions

After water restriction for 36 h, all the participants were in dehydration. Dehydration had negative effects on aspects of cognitive performance and mood, such as episodic memory, vigor, fatigue, esteem-related affect, and TMD. Water supplementation improved vigor, fatigue, TMD, processing speed, and working memory, but not episodic memory. A volume of 1000 mL, drunk by participants within 10 min, was the most appropriate to improve the hydration status, cognitive performance, and mood.

## Figures and Tables

**Figure 1 nutrients-13-03645-f001:**
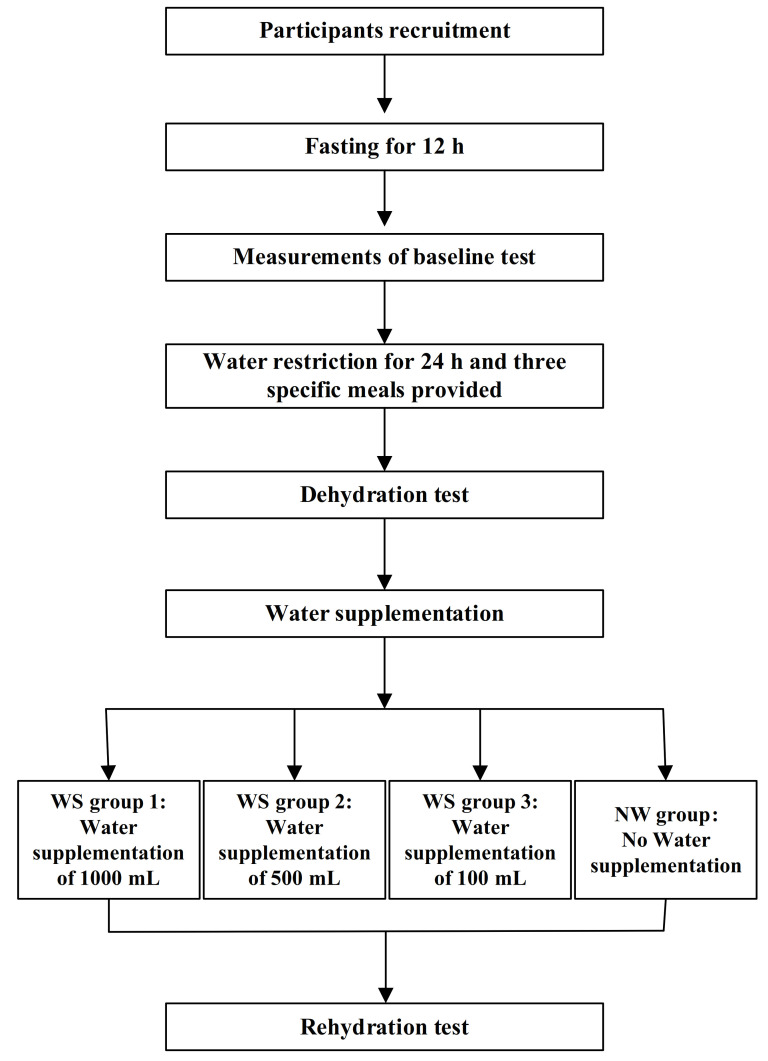
Study design.

**Figure 2 nutrients-13-03645-f002:**
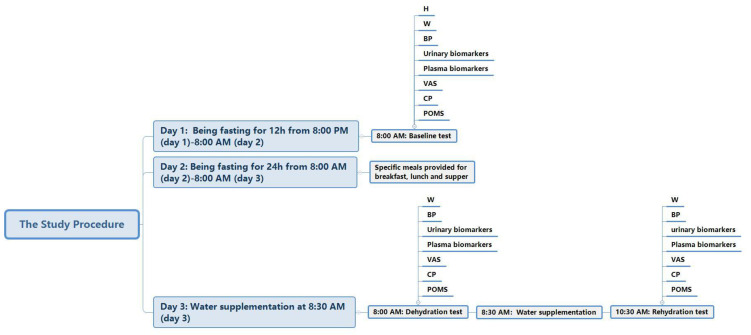
The study procedure (Note: H (height); W (weight); BP (blood pressure); VAS (visual analogue scales); CP (cognitive performance); POMS (profile of mood states).

**Table 1 nutrients-13-03645-t001:** The characteristics of the participants.

	Baseline Test					Dehydration Test					*t*	*p*	Rehydration Test				Interaction	
	WS Group 1 (*n* = 20)	WS Group 2 (*n* = 20)	WS Group 3 (*n* = 18)	NW Group (*n* = 18)	Total	WS Group 1 (*n* = 20)	WS Group 2 (*n* = 20)	WS Group 3 (*n* = 18)	NW Group (*n* = 18)	Total			WS Group 1 (*n* = 20)	WS Group 2 (*n* = 20)	WS Group 3 (*n* = 18)	NW Group (*n* = 18)	*F*	*p*
Age (year)	21 ± 1	21 ± 1	21 ± 1	21 ± 1	21 ± 1	21 ± 1	21 ± 1	21 ± 1	21 ± 1	21 ± 1	-	-	21 ± 1	21 ± 1	21 ± 1	21 ± 1	-	-
Height (cm)	168.3 ± 8.1	167.1 ± 7.8	163.5 ± 7.9	165.9 ± 9.0	166.3 ± 8.2	168.3 ± 8.1	167.1 ± 7.8	163.5 ± 7.9	165.9 ± 9.0	166.3 ± 8.2	-	-	168.3 ± 8.1	167.1 ± 7.8	163.5 ± 7.9	165.9 ± 9.0	-	-
Weight (kg) *	63.7 ± 9.0	64.4 ± 13.8	58.9 ± 11.9	60.8 ± 10.2	62.1 ± 11.4	63.4 ± 9.2	63.9 ± 13.8	58.5 ± 11.8	60.4 ± 10.2	61.6 ± 11.4	6.985	<0.001	63.9 ± 9.2	64.0 ± 13.8	58.5 ± 11.7	60.2 ± 10.1	1.102	0.354
BMI (kg/m^2^) *	22.6 ± 3.6	22.9 ± 3.8	21.9 ± 3.4	22.0 ± 2.3	22.4 ± 3.3	22.5 ± 3.6	22.7 ± 3.7	21.8 ± 3.3	21.8 ± 2.3	22.2 ± 3.3	7.162	<0.001	22.7 ± 3.7	22.8 ± 3.7	21.8 ± 3.3	21.8 ± 2.3	0.544	0.654
Systolic pressure (mmHg) *	112 ± 14	110 ± 12	109 ± 13	110 ± 11	110 ± 12	109 ± 10	107 ± 9	109 ± 11	107 ± 8	108 ± 10	3.142	0.002	113 ± 12	109 ± 10	110 ± 11	108 ± 11	0.842	0.475
Diastolic pressure (mmHg) *	67 ± 8	67 ± 11	63 ± 9	66 ± 7	65 ± 8	88 ± 7	86 ± 10	87 ± 9	86 ± 8	87 ± 8	30.874	<0.001	67 ± 8	64 ± 8	64 ± 9	65 ± 8	0.334	0.800

Note: Values are shown as the mean ± standard deviation (SD); *, there were significant differences between the baseline test and dehydration test.

**Table 2 nutrients-13-03645-t002:** The temperature and humidity of the study days.

	Indoors		Outdoors	
	Temperature (°C)	Humidity (%)	Temperature (°C)	Humidity (%)
First study day	20.7	59	22.0	64
Second study day	20.5	57	20.4	55
Third study day	20.5	63	19.8	61
Average	20.6	60	20.7	60

**Table 3 nutrients-13-03645-t003:** The water from food and 24 h urinary biomarkers of participants during the 24 h water restriction.

	WS Group 1 (*n* = 20)	WS Group 2 (*n* = 20)	WS Group 3 (*n* = 18)	NW Group (*n* = 18)	Total (*n* = 76)	*F*	*p*
Water from food	988 ± 253	870 ± 187	934 ± 210	884 ± 185	920 ± 212	1.264	0.293
Water from breakfast	79 ± 35	100 ± 40	82 ± 25	80 ± 26	85 ± 33	1.771	0.160
Water from lunch	463 ± 123	410 ± 99	448 ± 102	449 ± 101	442 ± 107	0.900	0.446
Water from supper	446 ± 130	361 ± 99	404 ± 122	355 ± 107	392 ± 119	2.568	0.055
Water from stable food	413 ± 146	367 ± 114	381 ± 111	352 ± 72	379 ± 115	1.001	0.398
Water from dishes	575 ± 167	503 ± 115	552 ± 137	532 ± 127	541 ± 138	0.965	0.414
24 h urinary biomarker							
Urine volume	939 ± 210	893 ± 187	934 ± 207	830 ± 136	900 ± 189	1.314	0.276
Voids	4 ± 1	4 ± 1	4 ± 1	4 ± 1	4 ± 1	0.949	0.422
24 h urine osmolality	973 ± 101	985 ± 104	948 ± 104	997 ± 83	976 ± 98	0.840	0.476
24 h-Na (mmol/L)	261 ± 35	247 ± 42	261 ± 33	267 ± 33	259 ± 36	1.015	0.391
24 h-K (mmol/L)	43.10 ± 10.99	45.27 ± 13.63	40.68 ± 9.71	51.08 ± 12.83	44.99 ± 12.28	2.523	0.064
24 h-Cl (mmol/L)	233 ± 30	213 ± 42	227 ± 31	223 ± 31	224 ± 34	1.165	0.329
24 h-Ca (mmol/L)	5.11 ± 2.28	5.18 ± 1.71	4.80 ± 1.56	4.75 ± 1.07	4.97 ± 1.71	0.303	0.823
24 h-P (mmol/L)	22.24 ± 6.26	26.99 ± 6.69	23.19 ± 5.79	25.42 ± 6.92	24.47 ± 6.58	2.194	0.096
24 h-Mg (mmol/L)	4.38 ± 2.08	4.30 ± 1.74	5.04 ± 2.41	5.08 ± 2.14	4.68 ± 2.08	0.753	0.524
24 h-pH	7.0 ± 0.4	6.9 ± 0.5	6.9 ± 0.3	7.1 ± 0.3	7.0 ± 0.4	1.735	0.167
24 h-USG	1.023 ± 0.005	1.025 ± 0.004	1.023 ± 0.005	1.022 ± 0.004	1.023 ± 0.004	1.733	0.168

Note: Values are shown as the mean ± standard deviation (SD).

## Data Availability

The data of this study is available from the corresponding author on reasonable request.

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
