# Peer review of "Effects of Water Restriction and Supplementation on Cognitive Performances and Mood among Young Adults in Baoding, China: A Randomized Controlled Trial (RCT)"

_nutrients, 2021, doi:10.3390/nu13103645_

Round 1
Reviewer 1 Report
Thank you for allowing me to read this manuscript. This is an interesting study and a well-written manuscript. Therefore, I only have some minor comments that the authors may wish to consider.
Abstract:
The abstract would benefit from the inclusion of a brief rationale and the aim made clearer. The inclusion of participant characteristics would be useful. Furthermore, stating the time scale for the ingestion of the 1000 ml would be a beneficial addition.
Introduction:
A clearer rationale for why China might be different from other countries, and thus require specific study would be helpful. Otherwise, the introduction provides a clear rationale for the study.
Methods:
Well described methods. Figure 1 is a helpful inclusion. It would be more logical to include characteristics of the participants in this section within the text.
Table 1: Report age as an integer.
Results:
Please consider including the actual P value for all outcomes.
It might make the results easier to understand if the plasma and urine results were presented in separate tables.
Discussion:
Lines 425 – 426: It is unclear why “Studies showed that the habitual caffeine consumption influenced the aspects of cognition, such as the attention and memory” is relevant to this section.
Lines 527 – 528: “The amount of 1000 ml was the most appropriate to improve the hydration status, cognitive performances and mood.” I feel that this needs to be developed in order to provide information about time scale as 1000 ml in a short period of time could be uncomfortable.
I hope that the authors find the above comments to be helpful and in the constructive manner in which they are intended.
Author Response
Reviewer 1
Open Review
(x) I would not like to sign my review report
( ) I would like to sign my review report
English language and style
( ) Extensive editing of English language and style required
( ) Moderate English changes required
(x) English language and style are fine/minor spell check required
( ) I don't feel qualified to judge about the English language and style
Yes |
Can be improved |
Must be improved |
Not applicable |
|
Does the introduction provide sufficient background and include all relevant references? |
(x) |
( ) |
( ) |
( ) |
Is the research design appropriate? |
(x) |
( ) |
( ) |
( ) |
Are the methods adequately described? |
(x) |
( ) |
( ) |
( ) |
Are the results clearly presented? |
(x) |
( ) |
( ) |
( ) |
Are the conclusions supported by the results? |
( ) |
( ) |
( ) |
( ) |
Comments and Suggestions for Authors
Thank you for allowing me to read this manuscript. This is an interesting study and a well-written manuscript. Therefore, I only have some minor comments that the authors may wish to consider.
Abstract:
The abstract would benefit from the inclusion of a brief rationale and the aim made clearer. The inclusion of participant characteristics would be useful. Furthermore, stating the time scale for the ingestion of the 1000 ml would be a beneficial addition.
Response: Thanks for your comments. It has been revised accordingly (Lines 17-27, Page 1). The brief, the inclusion of the participants and the time scale of the water supplementation of 1000 mL were added into the Abstract.
Introduction:
A clearer rationale for why China might be different from other countries, and thus require specific study would be helpful. Otherwise, the introduction provides a clear rationale for the study.
Response: Thanks for your comments. It has been revised accordingly in the Introduction Section (Lines 77-84, Page 2).
Only two large surveys about fluid intake were conducted. In 2009, Ma et al. conducted a fluid intake survey among 1483 adults from four cities in China, it was found that approximately 32% of the subjects drank less water than the 1200 mL/day recommended by the Chinese Nutrition Society. In another survey among 5868 primary and middle school students from the same four cities, it was reported that nearly two-thirds of the subjects drank less than the recommended amount. In addition, the results of the study conducted among young male adults showed that about 75% of them did not meet the recommendation of China in 2015 and 2017. The results of above-mentioned surveys imply that large proportion of Chinese residents may be in a state of dehydration, and their cognitive may be impaired. It would be more crucial for China to have more related studies to supply the scientific data, in order to improve the attention of Chinese to take in adequate water.
Studies demonstrated that the water intake were influenced by many factors including the age, gender, and temperature. This requires additional study in China due to the following reasons. The drinking patterns including the amounts and the types of the fluids intake differ among Chinese, Europeans, and Americans. The Chinese prefer tea, while Europeans and Americans tend to drink more coffee. In addition, the contributions of water from food were different among the countries. In Europe and the United States, the proportion of fluid intake from food accounts for about 20% of total fluid intake, whereas in China, the proportion of fluid intake from food accounts for nearly 44% or 50%. Furthermore, the landscapes vary significantly across its vast area, and the climates vary in different regions due to the highly complex topography in China. Heights are also different among the people of China and other countries due to race differences, which contributes to difference in body surface area and has corresponding effects on water requirements. Thus, although the studies related about the hydration status and cognitive performances among different countries had been conducted in some countries, it is still necessary for China to take corresponding studies to supply more information for the hydration status and cognitive performances.
Methods:
Well described methods. Figure 1 is a helpful inclusion. It would be more logical to include characteristics of the participants in this section within the text.
Table 1: Report age as an integer.
Response: Thanks for your comments. It has been revised accordingly in the Methods Section (Lines 128-129, Page 2).
The age of the participants was showed as an integer in Table 1.
Results:
Please consider including the actual P value for all outcomes.
Response: Thanks for your comments. We have made the revision accordingly (Lines 340-432, Pages 9-14).
It might make the results easier to understand if the plasma and urine results were presented in separate tables.
Response: Thanks for your comments. It has been revised accordingly, that the Table 4 was divided into Table 4 and Table 5 (Lines 391, 403, Pages 11-12).
Discussion:
Lines 425 – 426: It is unclear why “Studies showed that the habitual caffeine consumption influenced the aspects of cognition, such as the attention and memory” is relevant to this section.
Response: Thanks for your comments.
In this paragraph, we wanted to explain the characteristics that may affect the results of the cognitive performances, such as the gender, the caffeine consumption. The caffeine consumption may have a role in the hydration and cognitive performances. Furthermore, if the consumption of the caffeine was less than normal consumption occurred, then withdrawal effects could have influenced test results. Therefore, we explained the caffeine consumption of the participants that all the participants in our study did not have coffee drink habit.
Lines 527 – 528: “The amount of 1000 ml was the most appropriate to improve the hydration status, cognitive performances and mood.” I feel that this needs to be developed in order to provide information about time scale as 1000 ml in a short period of time could be uncomfortable.
Response: Thanks for your comments. It has been revised accordingly (Lines 556-557, Page 16).
All the participant that with 200, 500 and 1000 ml water supplementation were instructed to drink water within 10 minutes. Before the study protocol, we searched the articles about the effects of the amounts of the water supplementation, which varied from 150 ml to 1500 ml. Furthermore, in our previous study, the amount of the water supplementation was 1500 ml, and the participants were instructed to drink all the water within 15 minutes. After participants had the amount of the water supplementation of 1500 ml, we observed the behaviors of the participants, to record their uncomfortable phenomenon and we did not find any occur. Besides, in the present study, in the pre-experiment, all the researchers and a few participants took and completed the study protocols to find out the of the study. we recorded the feelings of the participants about the water supplementation and revised the details of the study protocols and we confirmed the amount and the time scale of the water supplementation of the study. Thirdly, the temperature of the water that supplied to the participants was maintained at 30℃-40℃, which was suitable for drinking, so as to reduce the gastrointestinal response of the participants. Finally, in order to avoid the discomfort of the participants during the study, all the researchers and experienced doctors from the emergency department and the urology department were 24h on call and all of them were asked to pay close attention to the participants before the study until 2 hours after the study in order to deal with emergencies.
I hope that the authors find the above comments to be helpful and in the constructive manner in which they are intended.
Response: Thanks for your comments.

Reviewer 2 Report
Here Zhang and colleagues conducted a randomized controlled trial within young adults in China to find that dehydration unexpectedly deteriorates episodic memory and mood. Specifically, they confirmed that water supplementation can greatly enhance processing speed, working memory and mood. The work is informative and innovative. However, there are some minor concerns to be addressed.
To begin with, the language usage requires formal editing and revision.
For example, “To investigate the effects of water restriction and supplementation on cognitive performances and mood and the optimum amount of water to alleviate the detriments of dehydration.” It is difficult to understand, and the sentence is not logic complete. It is highly recommended that the authors look into these misinformation and revise accordingly within the whole manuscript.
Methodologically, the authors should provide more details on the Cognitive performances (CP) test section. Are there any previous studies also applied this method and software that can be cited? From section 2.17 to 2.21, are the five tests these mentioned in 2.16? Or they are separate? All the cited references (33-35) in these sections are relatively not match with the procedures and methods for performance evaluations. The authors should double check and include more recent advanced methods to improve future interested readers.
In Table 3, the authors are welcome to explain how the water content from meals are calculated or measured? How to guarantee all the participants adhere to the guidelines to meet the end point of this dehydration/rehydration study?
Finally, dehydration/rehydration is reported to affect both acute and long term/working memory. The authors may add necessary discussions on this topic.
Collectively, the Reviewer recommends a Minor Revision to this manuscript.
Author Response
(x) I would not like to sign my review report
( ) I would like to sign my review report
English language and style
( ) Extensive editing of English language and style required
( ) Moderate English changes required
(x) English language and style are fine/minor spell check required
( ) I don't feel qualified to judge about the English language and style
Yes |
Can be improved |
Must be improved |
Not applicable |
|
Does the introduction provide sufficient background and include all relevant references? |
(x) |
( ) |
( ) |
( ) |
Is the research design appropriate? |
(x) |
( ) |
( ) |
( ) |
Are the methods adequately described? |
(x) |
( ) |
( ) |
( ) |
Are the results clearly presented? |
(x) |
( ) |
( ) |
( ) |
Are the conclusions supported by the results? |
(x) |
( ) |
( ) |
( ) |
Comments and Suggestions for Authors
Here Zhang and colleagues conducted a randomized controlled trial within young adults in China to find that dehydration unexpectedly deteriorates episodic memory and mood. Specifically, they confirmed that water supplementation can greatly enhance processing speed, working memory and mood. The work is informative and innovative. However, there are some minor concerns to be addressed.
To begin with, the language usage requires formal editing and revision.
For example, “To investigate the effects of water restriction and supplementation on cognitive performances and mood and the optimum amount of water to alleviate the detriments of dehydration.” It is difficult to understand, and the sentence is not logic complete. It is highly recommended that the authors look into these misinformation and revise accordingly within the whole manuscript.
Response: Thanks for your comments. We have made the revision accordingly in all the manuscript.
Methodologically, the authors should provide more details on the Cognitive performances (CP) test section. Are there any previous studies also applied this method and software that can be cited? From section 2.17 to 2.21, are the five tests these mentioned in 2.16? Or they are separate? All the cited references (33-35) in these sections are relatively not match with the procedures and methods for performance evaluations. The authors should double check and include more recent advanced methods to improve future interested readers.
Response: Thanks for your comments.
The cognitive tests were administrated using a “primary cognitive ability” software from Institute of Psychology, Chinese Academy Sciences, of which was the Deming Li in charge. The set of tests was sampled from 27 cities and counties in 20 provinces, completed the standardization work, and passed the appraisal of the professional committee of psychological measurement of the Chinese Psychological Society in May 2001. This set of tests was applicable to children, adolescents and the adults and elderly with education above the third grade of primary school. It has a wide application with the value in the aspects of clinic, rehabilitation, drug efficacy evaluation, talent selection and the effects of the education. The software was developed with the references of many classic and reliable literatures about the cognitive performances, which had been cited in our manuscript (from Section 2.17 to 2.21, references of 33-34). The literatures about the software were showed as follow, which were published in Chinese in many journals of China. But some data were not published yet because of the special characteristics of the people that took the tests. And the construction of the software was also cited in the study (Reference 33).
We have made the revision accordingly in the manuscript. Following are the references.
[1] Chen, T. Compilation of basic cognitive ability test for army personnel and report of measurement results. 2012.
[2] Chinese Academy Sciences. Investigation on important psychological characteristics of Chinese. 2017.
[3] Xu, S.; Wu, Z. The Construction of “The Clinical Memory Test”. Acta Psychol. Sin. 1986, 18, 102-110.
[4] Cai, X.; Huo, J.; Sun, J.; Huang, J.; Wang, J.; Wang, B.; Wei, Y. Effectiveness of iron fortified soy sauce on the cognitive ability of students [J].China Brewing, 2015, 4, 21-24.
[5] Cai, X.; Li, J.; Wei, Y.; Sun, J.; Huo J. Investigation on cognitive ability of middle school students from rural boarding school [J]. Chinese Journal of School Health, 2015, 36, 1179-1182.
[6] Cai, X. Investigation on cognitive ability and its influence factors of middle school students from rural boarding school [D]. 2015, Chinese Center for Disease Control and Prevention.
In Table 3, the authors are welcome to explain how the water content from meals are calculated or measured? How to guarantee all the participants adhere to the guidelines to meet the end point of this dehydration/rehydration study?
Response: Thanks for your comments.
Participants were asked to have the food with 75% water content that supplied by the researchers. All foods were weighed before and after the participants ate during the day 2. The backup food samples were collected and sent to laboratory of Beijing Institute Nutritional Resource to measure the water content according to National Food Safety Standard GB 5009.3–2016 Determination of water in Food.
There were several ways to maintain all the participants adhere to the protocols. Firstly, before taking part in the study, they were told clearly and signed the content inform and promised to take all the protocols and the researchers kept a close watch on the situation of the participants during the study. Secondly, the urine specific gravity, urine osmolality and plasma osmolality, provided sufficient power to confirm the hydration statuses of participants. Thirdly, we observed the situation of the participants’ mouth, in order to make sure that participants followed up the protocol of the study.
Finally, dehydration/rehydration is reported to affect both acute and long term/working memory. The authors may add necessary discussions on this topic.
Response: Thanks for your comments. We have made discussion in the manuscript (Lines 494-507, Page 15).
The episodic memory was one of the long term memory. In the present study, after water restriction of 36h, the scores of the episodic memory task decreased among the participants. But it did not improve after water supplementation, even with the maximum amount of 1000 ml. Therefore, it could be concluded that the water restriction of 36h could impede the episodic memory, but the 1000 ml may not be the amount that could attenuate the adverse effects. In the future, large volume of the water or longer intervention should be conducted.
Collectively, the Reviewer recommends a Minor Revision to this manuscript.
Response: Thanks for your comments.
